# Antibiotic Resistance and Genetic Determinants of *Helicobacter pylori* in Oman: Insights from Phenotypic and Whole-Genome Analysis

**DOI:** 10.3390/ijms26125628

**Published:** 2025-06-12

**Authors:** Amal Al-Hinai, Meher Rizvi, Said A. Al-Busafi, Masoud Kashoob, Zakariya Al-Muharrmi, Ahmed Al-Darmaki, Zaaima Al-Jabri

**Affiliations:** 1Department of Microbiology and Immunology, College of Medicine and Health Sciences, Sultan Qaboos University, Muscat 123, Oman; amhinai@squ.edu.om (A.A.-H.); rizvimeher@squ.edu.om (M.R.); 2Department of Medicine, College of Medicine and Health Sciences, Sultan Qaboos University, Muscat 123, Oman; busafis@squ.edu.om; 3Department of Medicine, Sultan Qaboos University Hospital, University Medical City, Muscat 123, Oman; kashoob@squ.edu.om; 4Microbiology and Immunology Diagnostic Laboratory, Department of Microbiology and Immunology, Sultan Qaboos University Hospital, University Medical City, Muscat 123, Oman; muharrmi@squ.edu.om; 5Department of Internal Medicine, Royal Hospital, Ministry of Health, Muscat 111, Oman; ahmed.aldarmaki@moh.gov.om

**Keywords:** *Helicobacter pylori*, antimicrobial resistance, virulence

## Abstract

*Helicobacter pylori* antibiotic resistance data in Oman are limited yet crucial for effective treatment selection. The genetic diversity within *H. pylori* influences its pathogenicity and clinical outcomes. This study evaluates resistance patterns and genetic determinants to guide treatment strategies. This study assessed antibiotic susceptibility in 15 *H. pylori* isolates (from 169 clinical samples) from naïve and treatment-failed patients. Resistance to clarithromycin (CLA), amoxicillin (AMX), metronidazole (MTZ), tetracycline, rifampicin (RIF), and levofloxacin (LEV) was tested alongside genetic analysis of virulence and resistance-associated mutations by whole-genome sequencing (WGS). Among the 15 resistant isolates, 20% were resistant to one antibiotic, 33.3% to two, 20% to three, and 26.6% to four antibiotics. MTZ resistance was universal among single-drug resistant isolates (100%). AMX-MTZ dual resistance was present in 60%, while triple resistance (CLA-AMX-MTZ) was present in 66.7%. Quadruple resistance (CLA-AMX-MTZ-RIF) was present in 75%. WGS revealed 23S rRNA mutations in 33.3% of CLA-resistant strains and *pbp-1* mutations in 66.6% of AMX-resistant strains. MTZ resistance was linked to *rdxA/frxA* mutations, while RIF and LEV resistance correlated with *rpoB* (65.7%) and *gyrA* (20%) mutations, respectively. The genotype–phenotype agreement was insignificant (*p* = 1). High mutation heterogeneity, virulence factors, and environmental influences contribute to resistance. Further studies on host–pathogen interactions are needed to understand resistance mechanisms.

## 1. Introduction

*Helicobacter pylori* (*H. pylori)* causes a range of complications that may vary from simple gastritis to chronic gastritis, peptic ulcers, as well as gastric cancer in the long term [1]. *H. pylori* infection, present in more than half of the global population and more common in rural areas, is often resistant to antibiotics and is associated with risk factors such as low socioeconomic status, a high sibling count, and maternal infection [2,3,4,5]. The current practice in treating *H. pylori* infection is based on a trial-and-error approach. Patients with confirmed infection are usually treated with different combinations of antibiotics and proton pump inhibitors (PPIs) [6,7]. While this approach has become a well-known practice, it is less efficient in treating the infection as resistance emerges. Antimicrobial stewardship has been implemented as a better systematic approach to improve efficacy and tackle resistance. This strategy includes selecting the most appropriate antibiotic dosage and duration of therapy, followed by properly monitoring patients’ compliance and response to treatment [8]. Despite extensive treatment efforts, *H. pylori* therapy often fails and has high recurrence rates. Research has shown that both bacterial virulence and host susceptibility are key factors in the infection’s pathogenesis, complicating the development of effective treatment strategies [9,10]. To improve eradication rates, many studies suggest that conducting antibiotic susceptibility testing and molecular detection before treatment could offer comparative advantages [11].

*H. pylori* can be eliminated by only a limited range of antimicrobial agents, yet resistance to these antibiotics is a growing global issue, complicating eradication efforts. While numerous studies have evaluated *H. pylori* antibiotic resistance worldwide, Oman lacks specific data. Hence, it is essential to examine the antimicrobial resistance trends of *H. pylori* in Oman to establish a local resistance profile in comparison with global trends. This study aims to provide a foundation for developing local clinical guidelines for diagnosing and treating *H. pylori* infections. This study’s primary objective is to explore the prevalence of *H. pylori* resistance to the six most common antibiotics used for treatment—clarithromycin, amoxicillin, levofloxacin, metronidazole, rifampicin, and tetracycline. Additionally, to the best of our knowledge, this is the first study from Oman to extensively perform genomic characterization of drug resistance and virulence factors in *H. pylori* local isolates through whole genome sequencing and to correlate these genetic characteristics with observed phenotypes.

## 2. Results

The antimicrobial susceptibility of 15 *H. pylori* isolates to six antibiotics was assessed using broth microdilution (BMD) and *E*-test methods (Table 1). For the comparison of antibiotic-resistant isolates between the two methods, a paired *t*-test, 95% confidence intervals (CIs), and *p*-values are shown in Table 2. For clarithromycin, 53.3% of isolates were resistant according to the BMD method, while 33.3% were resistant using the *E*-test. The *t*-test was −0.850, indicating that the sample mean scores for the BMD approach (mean = 18.0, SD = 37.6) and the *E*-test method (mean = 30.1, SD = 71.9) were statistically similar (*p* = 0.410) (Table 2).

The 95% CI agreement limits for the six antibiotics were as follows: (−42.6, 18.4) for clarithromycin (CLA), (−9.2, 28.4) for amoxicillin (AMX), (−94, −8.4) for tetracycline (TET), (−8.4, 12.6) for metronidazole (MTZ), (−6.5, 20.7) for rifampicin (RIF), and (−0.8, 1.6) for levofloxacin (LEV). In both the *E*-test and BMD methods, metronidazole showed the highest levels of resistance, with rates of 93.3% and 100%, respectively (Table 1). However, it was determined that the average scores for the isolates differed significantly between the BMD method (mean = 204.8, SD = 77.3) and the *E*-test method (mean = 256, SD = 0.0) (*p* = 0.022), despite both methods testing the same isolates.

When testing amoxicillin, resistance was found in 73.3% of isolates using broth microdilution and 66.7% using the *E*-test (Table 1). Further analysis indicated a *t*-test result of 1.096, with a *p*-value of 0.292, which is greater than 5%. This shows that the sample mean scores for the BMD method (mean = 38.6, SD = 75.6) and the *E*-test method (mean = 29, SD = 71) are statistically similar.

Only one isolate (6.7%) showed resistance to both tetracycline and levofloxacin with both methods. For levofloxacin, the t-statistic was 0.708, and there was no statistically significant difference between the sample mean scores for the BMD method (mean = 0.87, SD = 2.2) and the *E*-test method (mean = 0.48, SD = 1.5) (*p* = 0.491). Similarly, for tetracycline, the mean scores for the BMD method (mean = 5.0, SD = 16.4) and the *E*-test method (mean = 2.9, SD = 8.1) were statistically equivalent (*p* = 0.675), with a *t*-statistic of 0.428.

According to European Committee on Antimicrobial Susceptibility Testing (EUCAST) guidelines, where an MIC of ≤1 µg/mL indicates susceptibility to rifampicin, 46.7% of isolates were resistant through BMD and 33.3% through the *E*-test. The *t*-test result for rifampicin was 1.108, with a *p*-value of 0.287, which is greater than 5%, indicating that the sample mean scores for the BMD method (mean = 9.9, SD = 32.7) and the *E*-test method (mean = 2.9, SD = 8.1, *p* = 0.287) are variable. A recent study suggested a higher MIC breakpoint of ≤ 4 µg/mL for susceptibility, and using this standard, none of the isolates were phenotypically resistant to rifampin in either method [12,13].

To compare the mean and difference of the MICs between BMD and the *E*-test for CLA, AMX, TET, MTZ, RIF, and LEV, scatter plots (Bland–Altman plots) are displayed in Figure 1. Agreement was observed between the two methods for all antibiotics except metronidazole.

According to phenotypic methods (*E*-test and BMD), resistance to single antibiotic versus double, triple, and quadruple drug patterns is presented in Table 3. None of the isolates were sensitive to all six antibiotics. All isolates (100%) exhibited resistance to either one, two, three, or four antibiotics (20%, 33.3%, 20%, and 26.6%, respectively). The highest single antibiotic resistance was to metronidazole (3/3, 100%), while resistance to amoxicillin and metronidazole (3/3, 60%) was most common for double-drug resistance. The triple-drug resistance pattern (CLA + AMX + MTZ) occurred in 66.7% (2/3) of isolates, and the quadruple-drug resistance pattern (CLA + AMX + MTZ + RIF) was found in 75% (3/4) of cases.

The determination of MIC breakpoints was compared to the genotypes. Strains with amino-acid substitutions in the 23S rRNA gene (A2146G, A2147G, C1707T, and A2144G) were identified as resistant to CLA. Strains resistant to AMX had amino-acid substitutions in penicillin-binding proteins (*pbps*) such as *pbp1A:* (S595G in 17 out of 19, 89.5%), (G595S in 2 out of 19, 10.5%), and (S414R, N562Y, and T593A in 1 out of 19, 5.3%). For *pbp2,* amino-acid substitutions were found in (S494H and E572G), and for *pbp3,* (A541T and F490Y). Previous studies have reported similar amino-acid substitutions [14,15]. The common T556S mutation in *pbp1-A*, frequently linked to AMX resistance, was absent in this study. Resistant strains (HP3, HP5, and HP6) (MIC 64–128) and susceptible strains (HP1, 10, 14, and 15) showed different *pbp1-A* amino-acid substitutions, though all susceptible strains lacked T593A, which was present in only one resistant strain (HP3). Amino-acid substitutions in *pbp2* and *pbp3* did not appear to significantly influence AMX resistance.

Nineteen out of 20 strains were resistant to AMX, with one sensitive strain (HP6) (Figure 2). MTZ-resistant strains showed amino-acid substitutions in *rdxA* (T31E, R90K, A118T, A67V, C49T, R16C, D59N, H97T, H97Y, K64N, P106S, and S108A) and *frxA* (Y62D, A16T, A15V, A85V, M126F, A70G, and V7I). Additionally, amino-acid substitutions in *rpoB* (K2068R, I837V, and Q2079K) were linked to RIF resistance, while amino-acid substitutions in *gyrA* (N87K, D91G, and V172I) were associated with LEV resistance. No resistance-conferring genes were identified for TET.

A comparison of the two methods (genotypic and *E*-test) using the Mann–Whitney U test showed a score of 18 (*p* = 1), indicating no significant differences between the two methods for the six antibiotics (Table 4).

The analysis of genetic amino-acid substitutions was correlated to the resistance phenotype. Eleven strains were resistant to CLA, while nine were sensitive (Table 4). Resistant strains had amino-acid substitutions in the 23S rRNA gene at A2144G and C1707T (11 out of 11, 100%), A2147G (4 out of 11, 36.4%), and A2146G (3 out of 11, 27.1%) (Figure 2).

These amino-acid substitutions have been noted in previous studies [16,17,18,19]. However, this study did not find the commonly reported A2142C, A2142G, and A2143G amino-acid substitutions often associated with CLA resistance, suggesting possible geographic variation. Interestingly, the amino-acid substitutions identified (A2146G, A2147G, C1707T, and A2144G) did not appear to significantly contribute to CLA resistance, as they were present in both resistant (e.g., HP9, MIC > 128 mg/L) and susceptible strains (e.g., HP1, MIC 0.25 mg/L), while some resistant strains (e.g., HP6, MIC > 128 mg/L) lacked these amino-acid substitutions.

Metronidazole resistance was linked to a range of amino-acid substitutions in the *rdxA* and *frxA* genes, with significant variation among resistant strains. All 19 resistant strains had *rdxA* amino-acid substitutions, including the susceptible strain (HP13, MIC 8), though the positions varied. Common amino-acid substitutions included C49T (18 out of 20, 90%), D59N (17 out of 20, 85%), and others such as R90K, A118T, and H97T. *frxA* amino-acid substitutions included Y62D (13 out of 20, 65%) and others such as M126F, A16T, and A70G. These amino-acid substitutions have also been reported in previous studies [15,20].

All 20 isolates were susceptible to TET by *E*-test, consistent with the absence of amino-acid substitutions in the 16S rRNA gene. Among the 13 strains resistant to RIF, amino-acid substitutions were found in *ropB* at K2068R (8 out of 13, 61.5%), I837V (10 out of 13, 76.9%), and Q2079K (10 out of 13, 76.9%). Four strains had amino-acid substitutions in the *gyrA* gene at N87K (one out of four, 25%), D91G (two out of four, 50%), and V172I (three out of four, 75%). No amino-acid substitutions were found in *gyrB*. One strain (HP8) showed high RIF resistance but lacked amino-acid substitutions in *ropB*. Thus, amino-acid substitutions in *ropB* and *gyrA* may not fully account for RIF and LEV resistance. In some studies, the HefC protein has been linked to AMX but also cross-resistance to MTZ, and CLA by an efflux mechanism [21].

MLST analysis showed that, according to the CGE database (Appendix A), the sequence type of all isolates was classified as novel. However, based on the HpTT database, twelve isolates (HP1, HP2, HP3, HP5, HP8, HP9, HP13, HP14, HP15, HP16, HP17, and HP18) were identified as belonging to ST 3120. Additionally, one isolate (HP12) was classified as ST 3104, while seven isolates (HP4, HP6, HP7, HP10, HP11, HP19, and HP20) were determined to be novel.

Ninety genes which are linked to the virulence factors examined in this study are categorized based on the virulence factor database (VFDB) using *H. pylori* (ATCC 26695) as the reference genome (http://www.mgc.ac.cn/cgi-bin/VFs/v5/main.cgi accessed on 20 October 2021). The adherence virulence factor genes *(babA/hopS*, *babB/hopT*, *alpA/hopC*, *alpB/hopB*, *sabA/hopP*, *sabB/hopO*, *hpaA, hopZ*, and *horB*) were detected in some of the isolates (Figure 3). Specifically, *babA/hopS*, *hpaA*, *hopZ,* and *sabA/hopP* were only present in the strains. Additionally, the Lewis antigen, which is responsible for immune evasion and triggering autoimmune responses, was found in these strains with both *futB* and *futC* genes present but lacking *futA*. The lipopolysaccharide (LPS) gene *fucT* was detected in all strains except HP5 and HP20. Regarding immune modulator genes, only *napA* was identified in the strains. Analysis of flagellar genes (*fla*, *flg*, *flh*, and *fli*) revealed that most strains possessed these genes. All isolates contained *flaA* and *flaB*, but not *flaG*. Most strains also carried the *flg* (A, B, C, D, E, G, H, I, and K) genes, with the exception of *flgL*, and two copies of *flgG* and *flgE.* The strains had all the *flh* and *fli* genes listed in VFDB, including *flhA*, *flhB_1*, *flhB_2*, *flhF*, *fliA*, *fliD*, *fliE*, *fliF*, *fliG*, *fliH*, *fliI*, *fliL*, *fliM*, *fliN*, *Flip*, *fliQ*, *fliR*, *fliS*, *fliW2*, *fliW1*, and *fliY*.

For cytotoxins, 27 genes related to the Type IV secretory protein and the cag pathogenicity island (CagPAI) were found. All isolates contained *cag1*, *cag2*, *cag3*, *cag4*, *cag5*, *cagA*, *cagS*, *cagT*, and *virB11*, but none had *cagC*, *cagD*, *cagE*, *cagF*, *cagG*, *cagH*, *cagI*, *cagL*, *cagM*, *cagN*, *cagP*, *cagQ*, *cagU*, *cagV*, *cagW*, *cagX*, *cagY*, or *cagZ*. All isolates tested positive for VacA. Regarding the *dupA* and *iceA* genes associated with promoting duodenal and peptic ulcers, only one copy of the *iceA2* gene was present in the strains. *iceA1* (249 bp) and *iceA2* (390 bp) are allelic variants of *the iceA* gene (519 bp) with variable sizes [22].

Figure 4 shows a phylogenetic tree of the 20 local isolates from this study compared with the reference genome (strain ATCC 26695). The tree was generated using whole-genome (WG) SNPs with default parameters from the CSI phylogeny website and visualized through the iTol program. The alignment underwent a bootstrap pseudo-analysis with a value of 0.1. The isolates are grouped by their sequence types (STs) and are marked with different colors.

Group A (pink) represents ST 3120 strains (HP1, HP2, HP3, HP5, HP8, HP9, HP13, HP14, HP15, HP16, HP17, and HP18), group B (light yellow) includes ST 3104 (HP12), while group C (light green) contains novel STs (HP4, HP6, HP7, HP10, HP11, HP19, and HP20). Based on the region comparison with the reference genome using the HpTT tool, the analysis revealed that 15 of the 20 genome sequences (HP1, HP2, HP3, HP5, HP8, HP9, HP11, HP12, HP13, HP14, HP15, HP16, HP17, and HP18) clustered with the hpSouth America clade, one with hpEurope (HP4), one with Australia (HP10), and three with the hpAsia (India) clade (HP6, HP7, and HP19). None of the genomes were closely related to the hpAfrica clade.

To compare the lineages of our local strains from this study with global strains, a WG SNP-calling phylogenetic tree was constructed (Figure 5). These strains belong to international clones from Asia, including Iran and India, and Russia, while others were from Europe, such as Germany. Clustering of strains based on geographical location was observed in all clades. Strains from Oman are mainly found in two clusters; one of which is very close to the Indian clade, while two strains are more closely related to strains from Iran and could be allelic variants. Similar to the Omani local strains, ST-3120 was the most common ST among strains of *H. pylori* from Iran, Russia, and Germany. However, most of the strains from India had novel STs. Other miscellaneous STs were found scattered, including ST-3084, ST-3066, ST-3096, and, to a lesser extent, ST-3100 and ST-1233 only in one strain each.

## 3. Discussion

Multiple factors contribute to the rise of antibiotic resistance globally, including the overuse and misuse of antibiotics that are not in accordance with clinical guidelines, such as the widespread use of macrolides like azithromycin during the COVID-19 pandemic. Additional contributors include host-related factors, a lack of patient adherence, and pathogen-related factors such as amino-acid substitutions, virulence, and gene interactions [23]. In our *H. pylori* clinical isolates, resistance seems to be driven by factors beyond point mutations. The absence of strong statistical correlation implies that amino-acid substitutions alone may not be enough to accurately predict resistance, possibly due to other influences like gene expression, environmental conditions, or compensatory amino-acid substitutions.

Similar low isolation rates have been reported in other studies. For instance, only 26% of clinical strains were isolated at Cho Ray Phnom Penh Hospital in Southeast Asia [24], 53% at the University of Zurich [25], and 23% at Taleghani Hospital in Tehran [26]. Bacterial culture and phenotypic antimicrobial susceptibility testing (AST) are crucial for diagnosing *H. pylori* infection and determining its antibiotic susceptibility. Using conventional methods, AST results typically take around two weeks. Various in vitro methods have been reported for evaluating *H. pylori* antibiotic susceptibility, such as the *E*-test, agar dilution, and BMD. This study used both the *E*-test and BMD (equivalent to agar dilution), following methods from previous studies [27,28]. There were some variabilities observed between the BMD and *E*-test results in *H. pylori* susceptibility testing, which can be attributed to several well-documented factors. Methodologically, the *E*-test, a gradient diffusion technique, is more susceptible to influences such as agar composition, moisture, and inoculum density, while BMD offers more uniform antibiotic distribution in a liquid medium. These intrinsic differences may lead to inconsistencies, especially for antibiotics with narrow therapeutic windows or borderline MIC values. Additionally, the fastidious and patchy growth of *H. pylori* on solid media can compromise the clarity of *E*-test readings compared to the more controlled environment of BMD. The visual interpretation of *E*-tests may further introduce subjectivity, particularly for diffuse or trailing inhibition zones. To enhance reproducibility, all susceptibility tests in this study were conducted in duplicate on separate days, with a third repeat for any discrepancies exceeding one dilution; the consensus result was reported.

In Oman, the most common resistance patterns observed were for clarithromycin, amoxicillin, metronidazole, and rifampicin, with resistance rates from the *E*-test and BMD as follows: (CLA 53.3%, 55%), (AMX 73.3%, 66.7%), (MTZ 93.3%, 100%), and (RIF 46.7%, 65.7%), respectively. Resistance to tetracycline (6.7%, 0%) and levofloxacin (6.7%, 20%) was relatively low in both the *E*-test and BMD methods. These results align with those from recent studies [27,29]. Notably, there was no observed rifampicin resistance in other studies, which used a suggested MIC threshold of ≤4 µg/mL [12,13]. The lower MIC breakpoint for rifampicin (≤1 μg/mL) used in this study is consistent with the recent literature identifying low-level resistance linked to point mutations in the *rpoB* gene. This discrepancy from older thresholds (e.g., ≤4 μg/mL) may reflect regional genetic variability among *H. pylori* isolates or methodological differences across studies [20,26]. Ongoing molecular analyses of the *rpoB* gene are expected to clarify the genetic basis of this variability. Additionally, there were no significant correlations between the prevalence of antibiotic resistance and patient demographics, clinical results, or diagnoses, consistent with findings from prior studies [30].

WGS was employed to identify AMR-related amino-acid substitutions, revealing critical amino-acid substitutions linked to phenotypic resistance. For instance, CLA resistance (55%) was associated with A2144G, C1707T, A2147G, and A2146G amino-acid substitutions in the 23S rRNA gene, consistent with studies from China and Japan [18,31]. These amino-acid substitutions have also been reported in previous studies [20,24,32,33].

Amoxicillin resistance is commonly associated with amino-acid substitutions in penicillin-binding proteins (PBPs). In this study, the key *pbp1A* amino-acid substitutions were at S595G, G595S, N562Y, T593A, and S414R, with most isolates harboring the S595G mutation. This mutation has only been reported in one other study [34], while G595S, N562Y, and S414R have been detected in earlier research [24,35,36]. *pbp2* amino-acid substitutions at S494H and E572G were found in all strains except HP6, consistent with prior findings [14]. Amino-acid substitutions in *pbp3* at A541T and F490Y were also associated with AMX resistance, as documented in other studies [13,15]. It should be noted that not all *pbp1A* mutations confer functional resistance, and some may be naturally occurring polymorphisms with minimal impact on drug binding. Further structural or site-directed mutagenesis studies are needed to confirm their functional significance. However, despite the fact that the global prevalence of phenotypic AMX resistance remains low, our data shows a high rate of genotypic hits, and more phenotypic resistance could reflect the high use of amoxicillin in other common infections, including upper respiratory tract infections and urinary tract infections.

MTZ resistance, which has reached 100% in some regions, was observed in most isolates in this study. *rdxA* and *frxA* gene amino-acid substitutions were strongly linked to MTZ resistance, with *rdxA* amino-acid substitutions at T31E, R90K, A118T, and others, while *frxA* amino-acid substitutions were at Y62D, A16T, and A85V. These amino-acid substitutions have been observed in previous studies [20,34], suggesting a widespread resistance mechanism, partly attributed to the extensive use of metronidazole for various infections [23,37].

For TET, no amino-acid substitutions in the 16S rRNA gene were detected, aligning with findings from earlier research [35]. RIF resistance was linked to amino-acid substitutions in the *rpoB* gene at K2068R, I837V, and Q2079K, consistent with previous reports [15,38]. LEV resistance was observed in four strains, with *gyrA* amino-acid substitutions at N87K, D91G, and V172I, similar to earlier findings [13]. WGS aids in identifying amino-acid substitutions across various genes, facilitating the selection of appropriate antibiotics from the onset of therapy. The lack of significant association between certain drug resistance genes and phenotypic resistance in *H. pylori* may be explained by multiple factors beyond point mutations. While resistance to antibiotics such as clarithromycin and fluoroquinolones is typically mediated by mutations in 23S rRNA and gyrA, respectively, alternative mechanisms such as efflux pump overexpression (e.g., hefA, hefC, and hp0605), reduced membrane permeability, or compensatory mutations involved in stress responses and DNA repair may also contribute to resistance phenotypes. Additionally, heteroresistance—wherein subpopulations with differing resistance profiles coexist—can lead to discrepancies between genotypic and phenotypic outcomes if minority variants are not detected by standard assays. Moreover, the functional significance of novel or poorly characterized mutations may not be apparent without further validation, such as site-directed mutagenesis or transcriptomic studies. Variability in phenotypic AST methods, particularly near clinical breakpoints, can also contribute to these discrepancies. Future work incorporating functional genomics and efflux pump expression profiling will be critical to better correlate genotype with resistance phenotype.

There was an apparent reversal of resistance when combination antibiotics were used. For instance, the observed elimination of MTZ resistance in combination regimens (MTZ + LEV and AMX + MTZ + RIF) can be attributed to known synergistic interactions among these antibiotics, which may overcome low-level or partial resistance mechanisms. MTZ resistance in *H. pylori* is often due to impaired activation via nitroreductases (rdxA/frxA), and this resistance may be reversed or circumvented when combined with drugs like levofloxacin or rifampicin that exert bactericidal activity via independent mechanisms. These combinations may also enhance intracellular penetration and exert complementary pharmacokinetics that contribute to the observed enhanced efficacy [38].

According to the CGE database, the sequence types (STs) of the *H. pylori* strains in this study represent novel types [39]. Using the *Helicobacter pylori* Typing Tool (HpTT) database, it was found that twelve isolates belonged to ST 3120 (HP1, HP2, HP3, HP5, HP8, HP9, HP13, HP14, HP15, HP16, HP17, and HP18), seven isolates were novel types (HP4, HP6, HP7, HP10, HP11, HP19, and HP20), and one isolate (HP12) was identified as ST 3104. The STs differed between the two databases, suggesting that the isolates represent two distinct populations globally. Moreover, the sequenced isolates in this study likely possess unique alleles, with none matching previously recognized STs, thus potentially representing the Omani population, a pattern observed in a previous study from New York [13]. The identification of novel STs in our *H. pylori* isolates likely reflects both the underrepresentation of regional strains in global databases and the high genetic diversity of *H. pylori*, which is shaped by geographical and host-specific factors. As this is among the first genomic investigations of *H. pylori* strains from this region, the discovery of new STs is not unexpected and highlights the need for expanded surveillance and sequencing efforts in Middle Eastern populations.

WGS data was used to create a phylogenetic tree, comparing the 20 clinical strains to previously identified *H. pylori* STs [40]. Using the HpTT tool, the Omani strains were found to be paraphyletic, clustering with different regions: fifteen strains (HP1, HP2, HP3, HP5, HP8, HP9, HP11, HP12, HP13, HP14, HP15, HP16, HP17, and HP18) were grouped with the hpSouth America clade, one strain (HP4) with hpEurope, one (HP10) with Australia, and three (HP6, HP7, HP19) with hpAsia (India). None of the isolates were closely related to the hpAfrica clade. These findings suggest that host ethnicity might play a role in strain colonization, as observed in previous research in South American populations [41]. The phylogenetic clustering of our isolates with regionally diverse *H. pylori* strains may reflect ancestral recombination events or recent cross-continental transmission, possibly mediated by migration or historical trade routes. Tools such as STRUCTURE, fineSTRUCTURE, and BEAST could be utilized in future studies to investigate the genetic admixture and potential transmission patterns more precisely with a larger sample size.

WGS also facilitated the identification of various virulence genes. Most urease enzyme genes were present in the analyzed strains, which play a key role in stomach colonization by converting urea into ammonia, neutralizing stomach acid, and aiding bacterial survival [42]. Adherence genes, such as *babA*/*hopS, hpaA*, *hopZ,* and *sabA*/*hopP*, which are associated with peptic ulcers and gastritis, were identified. Although these genes have been linked to antigens in the gastroduodenal tract, this study found no direct correlation between the presence of these adhesin genes and a pathogenic phenotype, aligning with another research [43]. All isolates carried Lewis’s antigens, including FutB and FutC, which help evade host immune surveillance through molecular mimicry, promoting long-term gastric infections [44]. The *napA* gene, responsible for stimulating gastric inflammation by attracting neutrophils, was also present in all strains [45]. Regarding motility, 28 flagellar genes were detected, with all isolates containing essential genes such as *flgE, flhB,* and *fliF*, known to influence *H. pylori* motility [22]. Furthermore, the *vacA* toxin, a key player in *H. pylori* pathogenicity, was present in all isolates, with strong links to increased toxicity, gastric inflammation, and gastric cancer [46]. The *cagA* gene, also found in all isolates, showed no statistically significant correlation with clinical outcomes, similar to findings in Saudi Arabia and Sudan [47,48]. The *iceA2* gene, found in all strains, has been linked to non-ulcer dyspepsia and silent gastritis, though its precise function remains unclear [22,49]. This study highlights the widespread presence of AMR across different *H. pylori* lineages, with phenotypic resistance to clarithromycin, amoxicillin, metronidazole, tetracycline, rifampicin, and levofloxacin found among the identified strains. Cultivating *H. pylori* in vitro is notably difficult, particularly in liquid media. However, this study demonstrated its ability to grow in various nutrient-rich media, including brain heart infusion broth and Mueller–Hinton broth, supplemented with 5% fetal bovine serum. Solid media were enriched with 5% sheep blood. Out of the 169 *H. pylori* samples, only 23 (13.7%) successfully grew, while the remaining isolates failed to cultivate from gastric samples. The low retrieval of clinical strains is attributed to the fastidious nature of *H. pylori*, which requires specific growth conditions. Common challenges include the absence of microorganisms in biopsy samples, non-culturable coccoid forms, degradation during transport, stringent growth requirements, contamination by other bacteria, or prior antibiotic use by patients [50]. Additionally, the rapid urease test may yield inaccurate results, contributing to false positives or negatives [51]. We acknowledge the limitation of a small sample size in our study. However, it should be noted that *Helicobacter pylori* is a fastidious, slow-growing microbe under standard laboratory conditions; it requires specific conditions that are difficult to provide in routine culturing. This situation leads most clinical microbiology laboratories to prefer non-culture-based diagnostic methods such as the urea breath test, stool antigen testing, or histopathological examination over bacterial isolation. Furthermore, in the COVID-19 pandemic, receiving elective procedures like oesophagogastroduodenoscopy (OGD) was very limited. This led to fewer gastric biopsies sent for culture; thus, clinical isolates became unavailable. Many published reports have discussed the effect of the pandemic on gastrointestinal diagnostics and microbial surveillance. Despite the limited number of isolates, our study contributes valuable genomic and antimicrobial resistance data from a region with limited *H. pylori* surveillance. Future studies should aim to increase the sample size, particularly as access to endoscopic procedures becomes normalized, to strengthen the generalizability of the findings.

This study has several limitations. The small sample size (15 isolates) due to COVID-19 restrictions and challenges in culturing *H. pylori* may limit generalizability. Phenotypic susceptibility testing, while standardized, may not fully replicate in vivo conditions, and the absence of selective media caused the risk of contamination. Additionally, incomplete patient treatment histories hindered correlating resistance with prior antibiotic exposure. While WGS provided robust genetic insights, novel mutations without established phenotypic associations warrant further functional validation. These constraints emphasize the need for larger, multicenter studies and enhanced culturing techniques to refine Oman’s resistance landscape.

## 4. Materials and Methods

### 4.1. Ethics Statement

This study received ethical approval from the Medical Research Ethics Committee at the College of Medicine and Health Sciences, Sultan Qaboos University, under approval number 2502, dated 13 July 2021.

### 4.2. Patient Data

This research was conducted as a prospective multicenter study in three tertiary-care centers in Oman. Gastric biopsy samples were collected from patients referred for endoscopy at three major medical facilities: Sultan Qaboos University Hospital, Royal Hospital, and the Creativity Private Medical Center. The patients presented with a range of symptoms suggestive of gastroduodenal diseases, such as dyspepsia, chronic constipation, abdominal pain, bloating, nausea, vomiting (dark brown), melena (dark grey stool), epigastric pain, iron-deficiency anemia, a family history of gastric cancer, anorexia, and medication-resistant symptoms. All patients who agreed to participate were required to provide written informed consent. The collection period for gastric biopsies spanned from September 2021 to May 2022.

Epidemiological data, clinical history, and medication records were obtained from the patients’ hospital electronic records. All collected data were recorded in a password-protected Excel file. To ensure confidentiality, patient names were anonymized by replacing them with serial numbers.

### 4.3. Sample Size Calculation

In prior studies, *H. pylori* infection rates among adults in Asia ranged between 40% and 66.6%. Based on these findings, this study estimated an infection rate of 53.3% for Oman. Regional and global studies have shown antibiotic resistance rates varying between 12% and 59%. The calculated sample size for this study was 383 patients, assuming a 5% margin of error and 95% confidence interval. Due to the constraints posed by time, COVID-19 restrictions, and some patients refusing to participate, only 200 isolates were included in the study. Not all isolates survived because *H. pylori* is known for its strict growth requirements. We estimated that 50 samples would exhibit antimicrobial resistance.

The following equation was used to calculate the sample size:$$n=\frac{\left\{Z_{\left(1-\frac{\alpha }{2}\right)}^{2}\cdot P\left(1-P\right)\right\}}{d^{2}}$$
where

-(*n*) is the sample size,-(*Z*) represents the level of confidence (95% confidence, Z = 1.96),-(*P*) is the estimated infection rate (0.53 for Oman),-(*d*) is the margin of error (5%).

### 4.4. Inclusion and Exclusion Criteria

This study included adult patients exhibiting gastroduodenal symptoms and in possession of a positive Campylobacter-like organism (CLO) test for *H. pylori*. Participants had to abstain from proton pump inhibitors (PPIs) and antibiotics for at least two weeks prior to endoscopy and the commencement of any new therapy. Patients were excluded from the study if they had received antibiotics or PPIs within the two weeks preceding the endoscopy. Other exclusion criteria included patients who declined to participate, pregnant or lactating women, and patients with a negative CLO test or a positive COVID-19 diagnosis.

### 4.5. Sample Collection

Prior to the endoscopy procedure, all participants (*n* = 169) signed an informed consent form. Three gastric biopsy samples were taken from each patient: two from the antrum and one from the corpus (body) of the stomach. The biopsy for culture was placed in saline (0.85%) and cultured within two hours or stored in brain heart infusion broth (Oxoid, Hampshire, UK) supplemented with 20% glycerol and 5% fetal bovine serum (FBS) (Sigma-Aldrich, St. Louis, MI, USA) for delayed culturing. If immediate culturing was not possible, samples were stored at −80 °C for future use. Another biopsy was subjected to a rapid urease test (RUT) (AMA-Med Oy, Helsinki, Finland), and the third sample was placed in 10% formalin for histopathological examination. If the RUT was positive, it confirmed the presence of *H. pylori*, and the biopsy was sent to the microbiology laboratory for culture or was frozen at −80 °C. To minimize the risk of contamination, culture specimens were collected before histology samples.

### 4.6. Bacterial Isolates and Identification

*H. pylori* require selective growth conditions, including specific agar and atmospheric requirements. In the laboratory, samples were cultured on non-selective blood agar plates containing 5% sheep blood. Plates were incubated at 37 °C in a microaerobic atmosphere generator, which maintains conditions with 5% oxygen, 10% carbon dioxide, and 85% nitrogen. The initial incubation period lasted 3–7 days, with plates re-incubated for up to 14 days if no growth was observed. *H. pylori* was identified based on its characteristic morphology, Gram staining, and positive results for urease, oxidase, and catalase tests. Wet mounts were used to assess motility after incubating *H. pylori* cultures for 10 min in Mueller–Hinton broth (MHB) (Oxoid, Hampshire, UK) supplemented with FBS. Cultures were harvested, preserved using CryoBeads (Mast Diagnostics, Derby, UK), and stored at −80 °C for further analysis. Frozen isolates were later thawed and sub-cultured as needed.

### 4.7. Phenotypic Antimicrobial Susceptibility Testing

Of the 169 biopsies cultured, 15 *H. pylori* clinical strains were successfully identified. Antimicrobial susceptibility was determined using two phenotypic methods: the Epsilometer test (*E*-test) and the broth microdilution (BMD) method. These methods were employed to determine the minimum inhibitory concentration (MIC) of clarithromycin (CLA), amoxicillin (AMX), metronidazole (MTZ), tetracycline (TET), rifampicin (RIF), and levofloxacin (LEV).

### 4.8. Determination of MIC by E-Test

The *E*-test was conducted for clarithromycin, amoxicillin, metronidazole, tetracycline (range 0.016–256 mg/L), rifampicin, and levofloxacin (BioMérieux and Liofilchem, Nürtingen, Germany) (range 0.016–32 mg/L). A no. 3 McFarland suspension of *H. pylori* was prepared with a viable count between (10^8^) and (10^9^) CFU/mL. The suspension density was visually compared with McFarland standards to ensure accuracy. *H. pylori* isolates were inoculated onto MH agar plates containing 5% sheep blood, and the *E*-test strips were applied using sterile forceps. After incubation at 37 °C under microaerophilic conditions for 3–4 days, the MIC values were read. Interpretation followed the EUCAST guidelines, with MIC breakpoints as follows: >0.125 mg/L for AMX, >0.5 mg/L for CLA, >8 mg/L for MTZ, >1 mg/L for LEV, and >1 mg/L for TET. For RIF, a MIC of ≤4 µg/mL indicated sensitivity, based on previous studies [12,13].

### 4.9. Determination of MIC by Broth Microdilution Method (BMD)

This method involves serial two-fold dilutions in sterile 96-well microtiter plates, with growth indicated by turbidity or cell deposits at the bottom of the wells. The presence of growth in round-bottom wells appears as a button or pellet centered in the middle, while growth in flat-bottom wells can be scattered. MIC values were determined based on CLSI and EUCAST guidelines.

### 4.10. Preparation of Antibiotic Solution

Antibiotic stock solutions of known potency were obtained from Sigma-Aldrich (St. Louis, MI, USA). These solutions were dissolved and diluted according to CLSI guidelines. The stock solutions were frozen and thawed only once before discarding. Two-fold serial dilutions were prepared for each antibiotic solution.

### 4.11. Quality Control for Antibiotics

Since MIC values can be influenced by various conditions, quality control was crucial. Reference strains, such as *Klebsiella pneumoniae* ATCC 700603, *Staphylococcus aureus* ATCC 29213, *Escherichia coli* ATCC 25922, and *Bacteroides fragilis* ATCC 25285, were used as control organisms for testing the antibiotics amoxicillin, clarithromycin, levofloxacin, tetracycline, rifampicin, and metronidazole. The results from these control organisms ensured the accuracy of the MIC experiments.

### 4.12. Inoculum Preparation

Once the quality control results were satisfactory, *H. pylori* strains from patients were tested. The direct colony suspension method, known for its convenience with fastidious organisms, was employed to prepare the inoculum. The culture suspension’s turbidity was adjusted to a McFarland standard of 2.0, equivalent to a density of (10^7^) to (10^8^) CFU/mL. The 2.0 McFarland inoculum was further diluted to (5 × 10^5^) to (5 × 10^6^) CFU/mL.

### 4.13. Broth Microdilution Procedure

The antimicrobial susceptibility of *H. pylori* isolates was assessed by serial two-fold dilutions across 96-well microtiter plates. Antibiotic concentrations ranged from 128 µg/mL to 0.25 µg/mL, except for levofloxacin, which ranged from 128 µg/mL to 0.004 µg/mL. MHB containing 5% FBS (50 µL) was added to all wells, and a multi-channel pipette was used to transfer the antibiotic solution. Bacterial suspensions (50 µL) were added to the wells, except for the control columns. The plates were incubated for four days at 37 °C in a microaerophilic environment, and visual turbidity was assessed to determine microbial growth. The MIC was recorded as the lowest concentration of the antibiotic with no observable growth. Clinical breakpoints for antibiotic resistance were defined as follows: CLA > 0.5 mg/L, AMX > 0.125 mg/L, MTZ > 8 mg/L, TET > 1 mg/L, RIF > 1 mg/L, and LEV > 1 mg/L.

### 4.14. Genomic DNA Extraction

After the thawing of the *H. pylori* strains, the cultures were grown on blood agar plates containing 5% sheep blood for 3–5 days at 37 °C under microaerobic conditions. Following sufficient growth, the isolates were subcultured for DNA extraction and WGS. After culturing the strains and performing antimicrobial susceptibility testing, genomic DNA extraction was conducted using a QIAamp DNA Stool Mini Kit (QIAamp^®^ genomic DNA kit, Hilden, Germany) with slight modifications from the manufacturer’s instructions. Bacteria from three culture plates were harvested and suspended in 180 μL of Buffer ATL. The suspension was incubated at 37 °C for 30 min, then at 50 °C for an additional 10 min to ensure complete lysis of the bacteria. Proteinase K and RNase A (Thermo Fisher Scientific, Winsford, UK) were added to degrade proteins and RNA. Buffer AL and ethanol were subsequently added, and the sample was transferred to a QIAmp spin column. After a series of centrifugation and washing steps using AW1 and AW2 buffers, the DNA was eluted in Buffer EB. The extracted DNA was stored at −20 °C for further analysis.

### 4.15. DNA Quality and Quantity Assessment

The DNA quality was evaluated using a 1000 NanoDrop UV spectrophotometer (Thermo Fisher Scientific, Waltham, MA, USA), with an absorbance ratio of 260/280 nm between 1.8 and 2.0, which indicated pure DNA. Gel electrophoresis was performed to ensure there were no RNA or protein contaminants. A GeneRuler 1 kb DNA ladder was used to estimate the DNA fragment sizes.

### 4.16. Whole Genome Sequencing (WGS)

After extracting and quantifying the DNA, the samples were sent to MicrobesNG in the United Kingdom for next-generation sequencing using Illumina technology. The sequences were subsequently registered in the MicrobesNG database (https://microbesng.com/ accessed on 16 July 2022). The genome assembly and annotation of *H. pylori* were conducted by MicrobesNG in the United Kingdom. Once the assembled sequences were received, further analysis was performed. Sequences were compared with similar DNA sequences in the GenBank database from the National Center for Biotechnology Information (NCBI), using *H. pylori* 26695 as a reference genome strain.

### 4.17. Bioinformatic Analysis

The Artemis software (version 18.2.0) from the Sanger Institute was employed to visualize the *H. pylori* contigs in GFF format. The Multilocus Sequence Typing (MLST) program from the Center for Genetic Epidemiology in Denmark was used to perform MLST typing for all isolates (http://www.genomicepidemiology.org accessed on 4 November 2022). Additionally, the *H. pylori* Typing Tool (HpTT), (https://db.cngb.org/HPTT accessed on 15 November 2022) was used to assign geographic locations to the isolates. HpClass* was classified by previous studies [41,52,53,54,55,56]. For phylogenetic analysis based on single-nucleotide polymorphisms (SNPs), the CSI phylogeny tool was utilized, and the resulting data were used as input for the iTOL tool (https://itol.embl.de/ accessed on 18 November 2022) to build an unrooted phylogenetic tree. To predict antimicrobial resistance, the genome fasta files were uploaded to the Comprehensive Antibiotic Resistance (CARD) database (https://card.mcmaster.ca/ accessed on 15 November 2022). The Resistance Gene Identifier (RGI) program predicted resistance using nucleotide sequence database homology and SNP models. This analysis was performed on 20 *H. pylori* strains to identify amino-acid substitutions in the 23S rRNA, *pbp, rdxA, frxA*, 16S rRNA, *rpoB*, and *gyrA* genes, which are associated with resistance to clarithromycin, amoxicillin, metronidazole, tetracycline, rifampicin, and levofloxacin, respectively. The Artemis software (version 18.2.0) and local BLAST online tool (accessed on 17 November 2 (version 18.2.0) 022) searches were used to identify the nucleotide sequences corresponding to virulence factors. The genome sequences of the 20 *H. pylori* strains were deposited in the GenBank database under accession number AP023320. The virulence factors were classified into seven functional groups: ureases, adhesins, Lewis’s antigens, immune modulators, cytotoxins, flagellar genes, and plasticity zones.

### 4.18. Statistical Analysis

Once data collection was complete, statistical analysis was performed using the Statistical Package for Social Sciences (SPSS, version 28). Descriptive statistics were used to describe patient characteristics, such as age, gender, symptoms, and diagnosis. Antibiotic resistance rates were calculated as percentages. Paired *t*-tests were used to evaluate differences in mean MIC values between the *E*-test and broth microdilution methods. The Bland–Altman plot was employed to assess the agreement between these two methods. Additionally, the Mann–Whitney U test was used to assess differences between genotypic and phenotypic antimicrobial resistance patterns. A *p*-value of less than 0.05 was considered statistically significant.

## 5. Conclusions

This study is the first in Oman to use both phenotypic and genotypic characterization of *H. pylori* antimicrobial susceptibility. This study provides compelling local evidence that should prompt clinicians to reconsider the use of classical triple therapy for *H. pylori* eradication, particularly given the high resistance rates to clarithromycin and metronidazole. The data support a shift toward alternative first-line regimens, such as bismuth-based quadruple therapy or levofloxacin-containing therapy [57,58]. This recommendation aligns with updated regional and international guidelines, which now endorse bismuth-based therapy as the preferred first-line treatment in areas with known resistance patterns or where susceptibility testing is not routinely performed [52,53]. Future research is necessary for public health surveillance of *H. pylori*, monitoring its spread, and tracking antimicrobial resistance trends. WGS offers a more detailed analysis of resistance compared to traditional phenotypic methods. Future research should focus on expanding sample sizes across more Omani centers, increasing public awareness of antimicrobial resistance, and developing local clinical guidelines for diagnosing and treating *H. pylori* infections.

## Figures and Tables

**Figure 1 ijms-26-05628-f001:**
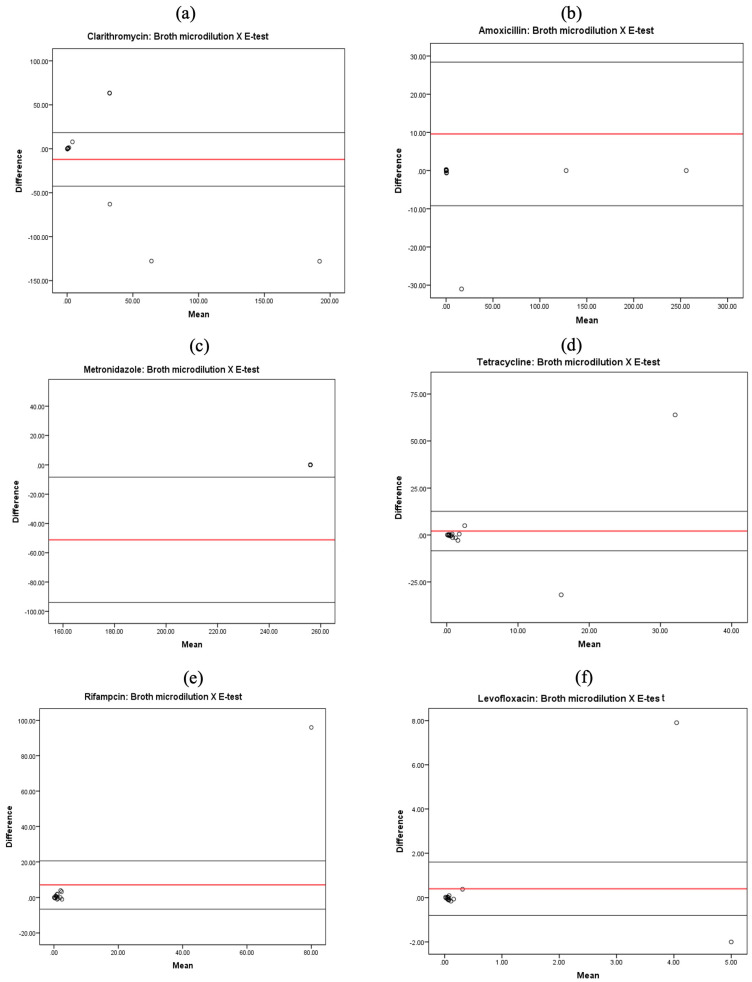
The minimum inhibitory concentration (MIC) results of the broth microdilution and *E*-test methods for the six antibiotics are shown in Bland–Altman difference plots (**a**): CLA, (**b**): AMX, (**c**): MTZ, (**d**): TET, (**e**): RIF, and (**f**): LEV. The MIC is plotted as empty circles in the figures.

**Figure 2 ijms-26-05628-f002:**
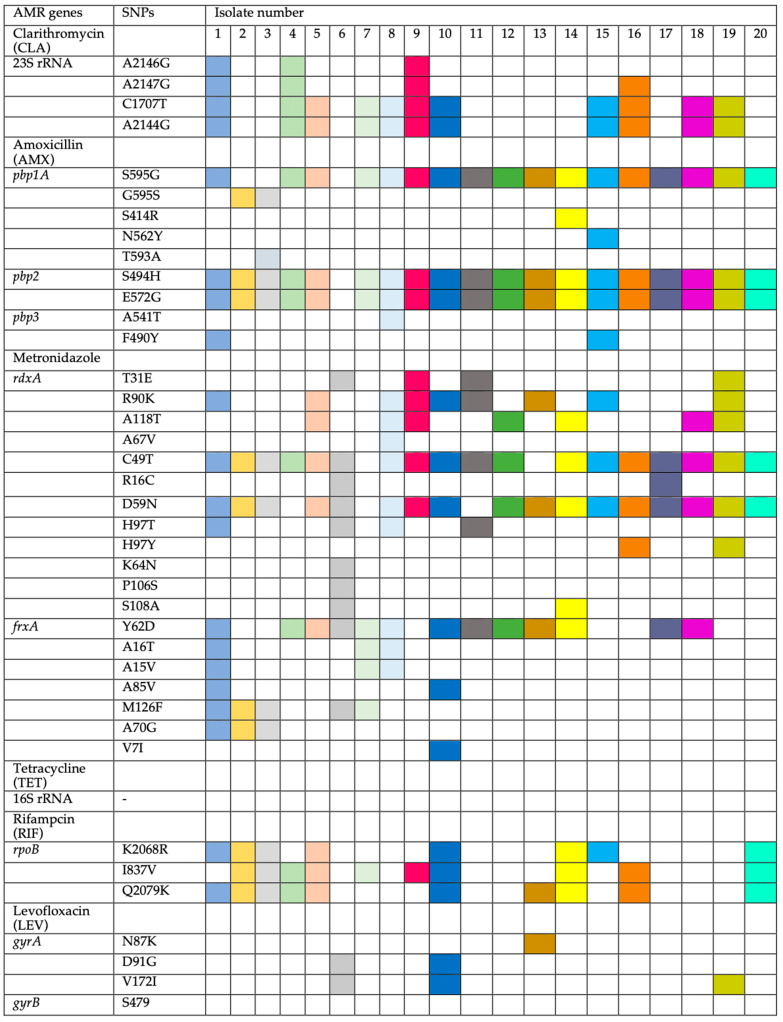
Amino-acid substitutions in 20 *H. pylori* isolates associated with resistance to CLA, AMX, MTZ, TET, RIF, and LEV. Amino-acid substitutions for each isolate is assigned with a different colour for visual clarification.

**Figure 3 ijms-26-05628-f003:**
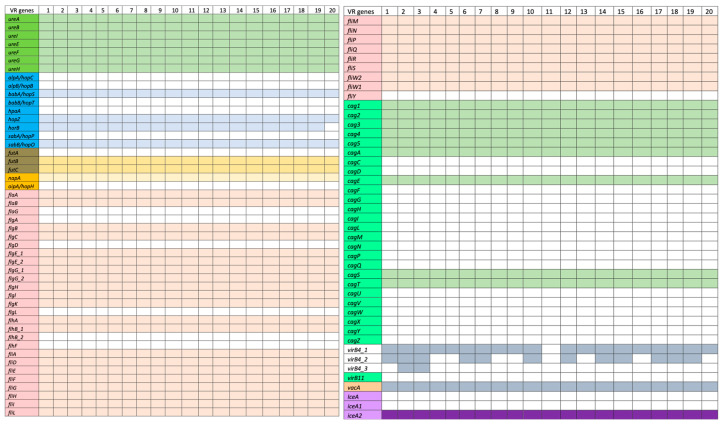
Classification of 90 *H. pylori* virulence-associated genes based on the virulence factor database (VFDB)**.** Each group of virulence genes are assigned with unique colours for visual clarification.

**Figure 4 ijms-26-05628-f004:**
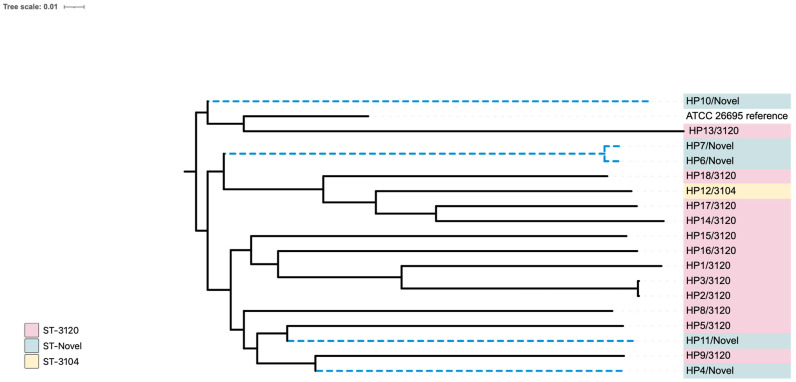
A phylogenetic tree of *H. pylori* local strains (*n* = 20): A maximum-likelihood tree demonstrating the genetic relatedness of different *H. pylori* strains based on WGS findings. The isolates are classified and color-coded according to their STs (pink = ST 3120, light yellow = ST 3104, light green = novel STs). The tree was visualized and annotated using iTOL.

**Figure 5 ijms-26-05628-f005:**
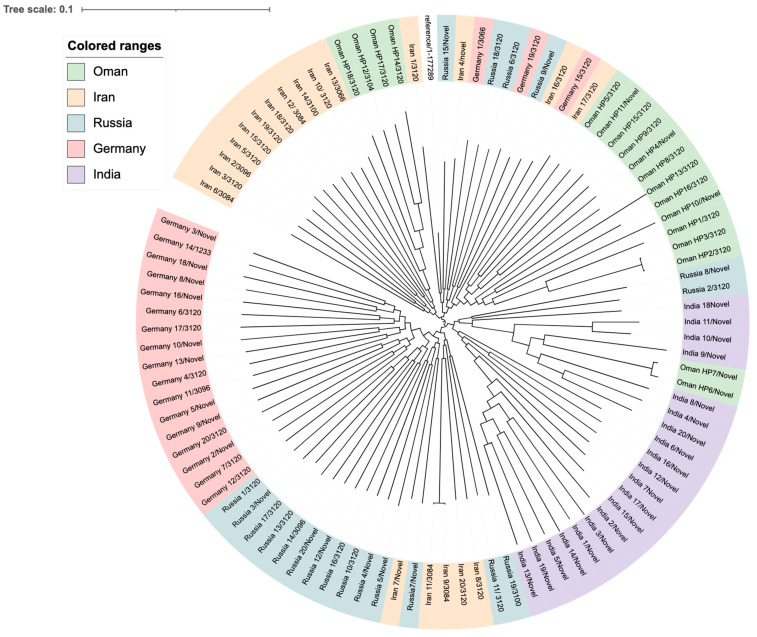
A whole-genome single-nucleotide polymorphism (SNP) calling phylogenetic tree of *H. pylori* isolates (*n* = 98). A maximum-likelihood tree showing the genetic relatedness between *H. pylori* strains based on WGS data. The strains are labeled with serial numbers and are grouped according to their STs. *H. pylori* ATCC26695 (GenBank accession number CP079087.1) was used as a reference. The isolates starting with the country name (in different colors) are globally published isolates retrieved from Genbank and were used for comparison. The isolates in green are strains from this study. The isolates in pink are from Europe (Germany), in blue are from Russia, in purple are from India, and in beige are from Iran. iTOL was used to visualize and annotate the tree. This figure compares the local strains in this study (represented in green) as well as global strains, which clearly demonstrates the diverse clonality of clinical isolates from Oman belonging to various STs.

**Table 1 ijms-26-05628-t001:** Comparison of antimicrobial susceptibility test results from broth microdilution and *E*-test method.

Method			Antimicrobial Agent					
			CLA ^1^	AMX ^2^	MTZ ^3^	TET ^4^	RIF ^5^	LEV ^6^
Broth microdilution		Range	128–0.125	128–0.125	128–0.125	128–0.125	128–0.125	128–0.032
		MIC (µg/mL)	0.25	0.125	8	1	1	1
	N (%) of isolates	Sensitive	7 (46.7)	4 (26.7)	1 (6.7)	13 (86.7)	8 (53.3)	13 (86.7)
		Resistant	8 (53.3)	11 (73.3)	14 (93.3)	2 (13.3)	7 (46.7)	2 (13.3)
*E*-test		Range	256–0.016	256–0.016	256–0.016	256–0.016	32–0.002	32–0.002
		MIC (µg/mL)	0.25	0.125	8	1	1	1
	N (%) of isolates	Sensitive	10 (66.7)	5 (33.3)	0 (0)	14 (93.3)	10 (66.7)	14 (93.3)
		Resistant	5 (33.3)	10 (66.7)	15 (100)	1 (6.7)	5 (33.3)	1 (6.7)

Abbreviations: ^1^ clarithromycin; ^2^ amoxicillin; ^3^ metronidazole; ^4^ tetracycline; ^5^ rifampicin; ^6^ levofloxacin.

**Table 2 ijms-26-05628-t002:** The paired *t*-test was employed in the broth microdilution method to compare the results of antimicrobial susceptibility testing from the *E*-test.

	Broth Microdilution		*E*-Test		*t*-Test	95% CI of the Difference		*p*-Value
	Mean	SD	Mean	SD		Lower	Upper	
CLA	18.0	37.6	30.1	71.9	−850	−42.6	18.4	0.410
AMX	38.6	75.6	29	71	1.096	−9.2	28.4	0.292
MTZ	204.8	77.3	256	0.0	−2.567	−94	−8.4	0.022
TET	5.0	16.4	2.9	2.9	0.428	−8.4	12.6	0.675
RIF	9.9	32.7	2.9	8.1	1.108	−6.5	20.7	0.287
LEV	0.87	2.2	0.48	1.5	0.708	−0.8	1.6	0.491

**Table 3 ijms-26-05628-t003:** Prevalence of resistance toward single vs. multiple drugs in *H. pylori* local isolates.

Susceptibility Test Result	N (%)
Number of strains	20
Sensitive	0
Single-drug resistance:	3 (20)
CLA ^1^	0
AMX ^2^	0
MTZ ^3^	3 (100)
TET ^4^	0
RIF ^5^	0
LEV ^6^	0
Double-drug resistance:	5 (33.3)
CLA + MTZ	1 (20)
MTZ + LEV	0 (0)
CLA + RIF	1 (20)
AMX + MTZ	3 (60)
Triple-drug resistance:	3 (20)
AMX + RIF + TET	1 (33.3)
AMX + MTZ + RIF	0 (0)
CLA + AMX + MTZ	2 (66.7)
Quadruple-drug resistance:	4 (26.7)
CLA + AMX + MTZ + RIF	3 (75)
AMX + MTZ+ RIF + LEV	1 (25)
Quintuple-drug resistance	0 (0)

Abbreviations: CLR ^1^, clarithromycin; AMX ^2^, amoxicillin; MTZ ^3^, metronidazole; TET ^4^: tetracycline, RIF ^5^, rifampicin; LEV ^6^, levofloxacin.

**Table 4 ijms-26-05628-t004:** Rates of resistance and susceptibility to six antibiotics according to detection methods (genotype and *E*-test)**.**

Method			Antimicrobial Drug					
			Clarithromycin	Amoxicillin	Metronidazole	Tetracycline	Rifampicin	Levofloxacin
Genotype (*n* = 20)	N (%) of isolates	Sensitive	9 (45)	1 (5)	0 (0)	20 (100)	7 (35)	16 (80)
		Resistant	11 (55)	19 (95)	20 (100)	0 (0)	13 (65.7)	4 (20)
*E*-test (*n* = 15)	N (%) of isolates	Sensitive	10 (66.7)	5 (33.3)	0 (0)	14 (93.3)	10 (66.7)	14 (93.3)
		Resistant	5 (33.3)	10 (66.7)	15 (100)	1 (6.7)	5 (33.3)	1 (6.7)

## Data Availability

All supporting data can be found in the Appendix A. All whole-genome sequencing data are deposited in DDBJ/ENA/GenBank under the accessions JAUQZG000000000-JAUQZX000000000. The version described in this paper is version JAUQZG000000000-JAUQZX000000000. The raw data supporting the conclusions of this article will be made available by the authors without undue reservation.

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
