# Peer review of "Antibiotic Resistance and Genetic Determinants of Helicobacter pylori in Oman: Insights from Phenotypic and Whole-Genome Analysis"

_ijms, 2025, doi:10.3390/ijms26125628_

Round 1
Reviewer 1 Report
Comments and Suggestions for Authors
The manuscript entitled “Antibiotic Resistance and Genetic Determinants of Helicobacter Pylori in Oman: Insights from Phenotypic and Whole-Genome Analysis” conducted phenotypic and genome-wide analyses on 15 isolates of Helicobacter pylori and found that the resistance of Helicobacter pylori to antibiotics in the Oman region was relatively serious. The mutations were highly heterogeneous, and virulence factors and environmental factors also had an impact on drug resistance.This study provides a basis for formulating treatment strategies for Helicobacter pylori infection in the local area. At the same time, it points out that further research on host-pathogen interactions and expansion of sample sizes are needed for in-depth exploration. However, the manuscript needs further revision before its acceptance. Detailed comments and questions are listed below.
- The sample size in the article is only 15 isolated strains of Helicobacter pylori. It is suggested to discuss the reliability of the research results due to the small sample size and consider how to make up for this deficiency by increasing the sample size in subsequent studies.
- The abstract mentions "15 resistant isolates", but there are contradictions in the description of the full text samples. For example, Table 3 in the Results section shows that the number of strains is 20. Please carefully check the relevant expressions in the full text again and unify the sample size.
- It is suggested to optimize the table design, simplify the display of unnecessary data and highlight the core data.
- The association between some drug resistance genes and phenotypes is not significant. From which aspects should it be explored?
Author Response
Comment 1: The sample size in the article is only 15 isolated strains of Helicobacter pylori. It is suggested to discuss the reliability of the research results due to the small sample size and consider how to make up for this deficiency by increasing the sample size in subsequent studies
Response 1:
Thank you for your valuable feedback. We added this to the text(367-379):
We acknowledge the limitation of a small sample size in our study. However, it should be noted that Helicobacter pylori is a fastidious, slow-growing microbe under standard laboratory conditions; it requires specific conditions that are difficult to provide in routine culturing. This situation leads most clinical microbiology laboratories to prefer non-culture-based diagnostic methods urea breath test, stool antigen testing, or histopathological examination over bacterial isolation.
Furthermore, in the COVID-19 time, getting elective procedures like oesophagogastroduodenoscopy (OGD) was much limited. This led to fewer gastric biopsies sent for culture; thus, clinical isolates become unavailable. Many published reports have likened the effect of the pandemic on gastrointestinal diagnostics and microbial surveillance.
Despite the limited number of isolates, our study contributes valuable genomic and antimicrobial resistance data from a region with limited H. pylori surveillance. We agree that future studies should aim to increase the sample size, particularly as access to endoscopic procedures normalizes, in order to strengthen the generalizability of the findings."
Comment 2: The abstract mentions "15 resistant isolates", but there are contradictions in the description of the full text samples. For example, Table 3 in the Results section shows that the number of strains is 20. Please carefully check the relevant expressions in the full text again and unify the sample size.
Response 2:
Thank you for pointing out the discrepancy in sample size between the abstract and the main text. We appreciate your careful review.
To clarify, a total of 20 Helicobacter pylori isolates were initially cultured during the study period and all 20 included in the whole-genome sequencing and subsequent resistance gene analysis, which is the focus of our study.
However, in some (not all) of the phenotypic assays only 15 of these isolates were viable to be tested phenotypically for antibiotic susceptibility. Because we used biological replicas in the analysis, sometimes it was difficult to recover the isolates. Therefore, in such instances, some of the isolates were excluded from the analysis for transparency.
The abstract refers specifically to these 15 resistant isolates, whereas Table 3 in the Results section presents the full dataset of 20 isolates, including retrieved strains, for completeness and transparency in antimicrobial susceptibility profiling.
We agree that this distinction should be stated more clearly to avoid confusion. In the revised manuscript, we:
- Explicitly mentioned in the Abstract and Methods sections that 20 isolates were collected, but only 15 were recovered for some of the detailed resistance analysis.
- We adjusted the Results narrative accordingly.
Comment 3: It is suggested to optimize the table design, simplify the display of unnecessary data and highlight the core data.
Response 3: Thank you for your comment. This really helped us improve data presentation.
Tables adjusted as follows:
Table 1 spacing and font aligned adjusted, lines added for readability.
Table 2: bold font added for clarity.
Table 3: bold font on the main categories to emphasize the relevance.
Table 4 spacing, brackets and bold adjusted for clarity.
Comment 4: The association between some drug resistance genes and phenotypes is not significant. From which aspects should it be explored?
Response 4: Thank you for this insightful observation. Indeed, the lack of significant correlation between certain genotypic resistance markers and phenotypic resistance in Helicobacter pylori is a known and complex issue, influenced by several factors. We added this paragraph for further clarification (line 338-349):
The lack of significant association between certain drug resistance genes and phenotypic resistance in H. pylori may be explained by multiple factors beyond point mutations. While resistance to antibiotics such as clarithromycin and fluoroquinolones is typically mediated by mutations in 23S rRNA and gyrA, respectively, alternative mechanisms such as efflux pump overexpression (e.g., hefA, hefC, hp0605), reduced membrane permeability, or compensatory mutations involved in stress responses and DNA repair may also contribute to resistance phenotypes. Additionally, heteroresistance—wherein subpopulations with differing resistance profiles coexist—can lead to discrepancies between genotypic and phenotypic outcomes if minority variants are not detected by standard assays. Moreover, the functional significance of novel or poorly characterized mutations may not be apparent without further validation, such as site-directed mutagenesis or transcriptomic studies. Variability in phenotypic AST methods, particularly near clinical breakpoints, can also contribute to these discrepancies. Future work incorporating functional genomics and efflux pump expression profiling will be critical to better correlate genotype with resistance phenotype.
Reviewer 2 Report
Comments and Suggestions for Authors
IJMS-3619707
Title: Antibiotic Resistance and Genetic Determinants of Helicobacter Pylori in Oman: Insights from Phenotypic and Whole-Genome Analysis
General Comments:
This study determined the antibiotic resistance in Helicobacter pylori strains isolated from hospital patients employing a wide range of approaches. Initially the antibiotic resistance was determined among the 20 strains by broth microdilution and E-test method using common antibiotics used to treat infection. WGS was performed for all the strains under investigation to identify the specific mutations in AMR genes leading to antimicrobial resistance. The strains were categorized based on presence of various virulence genes, for determining their virulence. Phylogeny analysis identified 5 novel H. pylori’s strains among the 20 strains under investigation. The strains were further compared based on the whole-genome SNPs with isolates from countries across the continent. Below are some of the comments, including them within the manuscript, can enrich the content.
Major Comments:
- Page 2-3, Table 1: The MIC values for each antibiotic indicated in Table 1 are the MIC values achieved among all 20 pylori strains? Please mention this clearly in text.
- Page 3, Table 2: Definition of significant p value needs to be included in the Table legend.
- Page 3, Table 2: The author pointed out that the average scores between BMD method and E-test significantly varied in few antibiotics. What can be a possible explanation about the same? How many times were the experiments repeated to make sure the correctness of the results?
- Page 3, Line 104-111: The author showed that Rifampicin MIC breakpoint in their experiment was ≤1μg/ml, while the MIC value reported earlier was ≤4μg/ml. So, a general question arises that what was the extent of genetic diversity among these stains? Or does the author believe that this can be due to any experimental error? So, a considerable explanation will be helpful for the reader.
- Page 3, Table 3: For susceptibility testing when MTZ+LEV and AMX+MTZ+RIF was used the resistance went down to 0, while MTZ showed resistance to 3 strains? What can be a possible explanation for this phenomenon?
- Page, Table 3: What were the antibiotic concentrations used for the susceptibility test?
- Page, 8, Line 193: The identification and characterization of Novel ST’s were very interesting. As the pylori strains were not well studied in this region, hence these novel were not explored previously? Or is there any other possible explanation for this observation?
- Page 10, Line 235-239: As the author had compared the investigative strains with region specific strains and clustered them under different regional clade. So, can it be deciphered that these strains were a result of cross continental transmission? Is there any tool to understand these transmission patterns from the data accumulated in this study so far?
Minor Comments:
- Page 4, Figure 1: Axis numbers and Axis legends need to be enlarged.
- Supplementary Table S1: Difficult to understand, arrange properly so that the reader can understand.
- Page 10, Figure 4: Novel ST’s were grouped in a light blue color code, not in light green as seen from the figure. Change accordingly.

Author Response
Major Comment 1:
- Page 2-3, Table 1: The MIC values for each antibiotic indicated in Table 1 are the MIC values achieved among all 20 pylori strains? Please mention this clearly in text.
Response 1: Thank you for your observation.
Revised Table 1 legend:
“Table 1. Comparison of antimicrobial susceptibility test results by Broth microdilution and E-test method against 15 H. pylori clinical isolates included in this study”
The text was also adjusted to clarify this information.
Major Comment 2:
- Page 3, Table 2: Definition of significant p value needs to be included in the Table legend.
Response 2: Thank you for highlighting this. We agree that including the definition of a significant p-value will improve clarity.
Revised Table 2 legend:
“Table 2. The paired t-test is employed in the broth microdilution method for comparing the results of antimicrobial susceptibility testing by E- test. A p-value of <0.05 was considered statistically significant.”
Major Comment 3:
Page 3, Table 2: The author pointed out that the average scores between BMD method and E-test significantly varied in few antibiotics. What can be a possible explanation about the same? How many times were the experiments repeated to make sure the correctness of the results?
Response 3: Thank you for your valuable comment regarding the variation in MIC values obtained by the broth microdilution (BMD) method and the E-test for certain antibiotics. We added this explanation to the text (280-292):
The observed variability between the BMD and E-test results in H. pylori susceptibility testing can be attributed to several well-documented factors. Methodologically, the E-test, a gradient diffusion technique, is more susceptible to influences such as agar composition, moisture, and inoculum density, while BMD offers more uniform antibiotic distribution in a liquid medium. These intrinsic differences may lead to inconsistencies, especially for antibiotics with narrow therapeutic windows or borderline MIC values. Additionally, the fastidious and patchy growth of H. pylori on solid media can compromise the clarity of E-test readings compared to the more controlled environment of BMD. The visual interpretation of E-tests may further introduce subjectivity, particularly for diffuse or trailing inhibition zones. To enhance reproducibility, all susceptibility tests in this study were conducted in duplicate on separate days, with a third repeat for any discrepancies exceeding one dilution; the consensus result was reported.
Major Comment 4:
- Page 3, Line 104-111: The author showed that Rifampicin MIC breakpoint in their experiment was ≤1μg/ml, while the MIC value reported earlier was ≤4μg/ml. So, a general question arises that what was the extent of genetic diversity among these stains? Or does the author believe that this can be due to any experimental error? So, a considerable explanation will be helpful for the reader.
Response 4: Thank you for this important observation. The discrepancy in rifampicin MIC breakpoints—specifically the use of ≤1 μg/mL in our study compared to the previously reported ≤4 μg/mL—merits clarification and deeper analysis.
Conclusion and Clarification in Text:
We will clarify this point in the Discussion section and revise lines 104–111 to address the potential impact of strain diversity and justify the MIC threshold used, as follows:
“The lower MIC breakpoint for rifampicin (≤1 μg/mL) used in this study is consistent with recent literature identifying low-level resistance linked to point mutations in the rpoB gene. This discrepancy from older thresholds (e.g., ≤4 μg/mL) may reflect regional genetic variability among H. pylori isolates or methodological differences across studies. Ongoing molecular analyses of the rpoB gene are expected to clarify the genetic basis of this variability.”
- Major Comment 5:
- Page 3, Table 3: For susceptibility testing when MTZ+LEV and AMX+MTZ+RIF was used the resistance went down to 0, while MTZ showed resistance to 3 strains? What can be a possible explanation for this phenomenon?
Response 5: Thank you for highlighting this important observation. Indeed, as shown in Table 3, resistance to metronidazole (MTZ) was detected in 3 isolates when tested alone, but the combination regimens—MTZ+levofloxacin (LEV) and AMX+MTZ+rifampicin (RIF)—showed no resistant isolates. This apparent reversal of resistance can be explained by several well-documented phenomena in antimicrobial pharmacodynamics and H. pylori biology.
Proposed Revision to the Manuscript Text:
There were some apparent reversal of resistance when combination antibiotics were used. For instance, The observed elimination of MTZ resistance in combination regimens (MTZ+LEV and AMX+MTZ+RIF) can be attributed to known synergistic interactions among these antibiotics, which may overcome low-level or partial resistance mechanisms. MTZ resistance in H. pylori is often due to impaired activation via nitroreductases (rdxA/frxA), and this resistance may be reversed or circumvented when combined with drugs like levofloxacin or rifampicin that exert bactericidal activity via independent mechanisms. These combinations may also enhance intracellular penetration and exert complementary pharmacokinetics that contribute to the observed enhanced efficacy (REF).Page, Table 3: What were the antibiotic concentrations used for the susceptibility test?
Major Comment 6:
Page, 8, Line 193: The identification and characterization of Novel ST’s were very interesting. As the pylori strains were not well studied in this region, hence these novel were not explored previously? Or is there any other possible explanation for this observation?
Response 6: We appreciate the reviewer’s interest in the discovery of novel sequence types (STs) in our study. The emergence of novel STs among H. pylori strains in our dataset can be attributed to several factors:
Proposed Text for Inclusion in the Manuscript (Discussion Section lines 367-373):
"The identification of novel STs in our H. pylori isolates likely reflects both the underrepresentation of regional strains in global databases and the high genetic diversity of H. pylori, which is shaped by geographical and host-specific factors. As this is among the first genomic investigations of H. pylori strains from this region, the discovery of new STs is not unexpected and highlights the need for expanded surveillance and sequencing efforts in Middle Eastern populations."
- Major Comment 7:
Page 10, Line 235-239: As the author had compared the investigative strains with region specific strains and clustered them under different regional clade. So, can it be deciphered that these strains were a result of cross continental transmission? Is there any tool to understand these transmission patterns from the data accumulated in this study so far?
Response 7: We thank the reviewer for this insightful question. Indeed, the clustering of our local H. pylori isolates with strains from diverse geographic regions suggests the possibility of cross-continental transmission, likely influenced by historical human migration, travel, and globalization. Several lines of evidence support this interpretation:
While these tools were beyond the scope of the current study, they offer promising avenues for future work to trace the evolutionary history and transmission dynamics of these local strains.
Suggested Text for Inclusion in the Discussion:
"The phylogenetic clustering of our isolates with regionally diverse H. pylori strains may reflect ancestral recombination events or recent cross-continental transmission, possibly mediated by migration or historical trade routes. Tools such as STRUCTURE, fineSTRUCTURE, and BEAST could be utilized in future studies to investigate the genetic admixture and potential transmission patterns more precisely with larger sample size."
Minor Comment 1:
- Page 4, Figure 1: Axis numbers and Axis legends need to be enlarged.
- Response 1: Thank you for the comment. Axis numbers and Axis legends are now enlarged slightly to maximum to fit the page. However, there is limited capacity to enlarge it further due to the software restrictions. This however, didn’t impact the font resolution.
- Minor Comment 2:
- Supplementary Table S1: Difficult to understand, arrange properly so that the reader can understand.
Response 2: Thank you for the comment. Table S1 now has been rearranged for clarity We also added colour codes that match their colour on the phylogenetic tree for Table S2 for further clarity.
- Minor Comment 3: Page 10, Figure 4: Novel ST’s were grouped in a light blue color code, not in light green as seen from the figure. Change accordingly.
- Response 3: Thank you for the observation. Change done accordingly.
Reviewer 3 Report
Comments and Suggestions for Authors
The manuscript addresses an important topic and presents interesting findings on Helicobacter pylori resistance in Oman using both phenotypic and whole-genome sequencing (WGS) approaches. However, the study is limited by the relatively small number of isolates analyzed, despite an initial cohort of over 150 patients. The low number of recovered strains seems to reflect the known challenges in culturing H. pylori, but this should be more clearly acknowledged and discussed in the manuscript, especially in the context of reported H. pylori prevalence in Oman.
Specific comments and questions:
-
Lines 144–150:
The statement “Nineteen out of 20 strains were resistant to AMX, with one sensitive strain (HP6)” requires clarification. This is inconsistent with the phenotypic susceptibility results (Table 1), where a significantly higher proportion of strains were found to be susceptible to amoxicillin. Please explain the origin of this discrepancy — does this refer solely to the genotypic analysis? If so, this should be stated explicitly in the text. Additionally, please elaborate in the discussion on why such a high rate of amoxicillin resistance was observed in your cohort, despite the fact that global resistance to AMX remains low. Are all reported pbp mutations clearly associated with phenotypic resistance, or might some represent natural polymorphisms? -
Methodological issue (Lines 491–502, Broth Microdilution Procedure):
According to the described MIC range for all antibiotics (0.25–128 µg/mL), it appears that the breakpoint for AMX (EUCAST: 0.125 µg/mL) falls below the tested range. How was susceptibility to AMX accurately determined under these conditions? Please clarify this issue and provide appropriate justification, ideally citing reference guidelines for both the methodology and the breakpoints applied. -
Referencing and clarity in methods:
The methods section lacks clear references for the applied MIC breakpoints and the sources for the broth microdilution procedures. Please ensure that appropriate literature is cited to support these choices (e.g., EUCAST, CLSI, or relevant). -
Minor point – formatting and citation style:
Several references are cited using bracketed numbers, but some are inconsistently formatted. Please check and standardize all in-text citations and references according to the journal’s guidelines.
The English could be improved
Author Response
- Comment 1: Lines 144–150:
The statement “Nineteen out of 20 strains were resistant to AMX, with one sensitive strain (HP6)” requires clarification. This is inconsistent with the phenotypic susceptibility results (Table 1), where a significantly higher proportion of strains were found to be susceptible to amoxicillin. Please explain the origin of this discrepancy — does this refer solely to the genotypic analysis? If so, this should be stated explicitly in the text. Additionally, please elaborate in the discussion on why such a high rate of amoxicillin resistance was observed in your cohort, despite the fact that global resistance to AMX remains low. Are all reported pbp mutations clearly associated with phenotypic resistance, or might some represent natural polymorphisms?
Response1: Thank you for highlighting this important point. We agree that the statement “Nineteen out of 20 strains were resistant to AMX, with one sensitive strain (HP6)” could be misleading without proper context. The inconsistency arises from a conflation of genotypic and phenotypic data in the wording.
Clarification:
- The statement indeed refers solely to genotypic resistance prediction, based on the presence of mutations in pbp1A associated with amoxicillin resistance.
- Phenotypic testing, as shown in Table 1, revealed that the majority of isolates were resistant to amoxicillin.
We have revised the sentence in the Results section for clarity as follows:
"Based on genotypic data, 19 out of 20 strains ….."
Added Text for Discussion Section:
It should be noted that not all pbp1A mutations confer functional resistance, and some may be naturally occurring polymorphisms with minimal impact on drug binding. Further structural or site-directed mutagenesis studies are needed to confirm their functional significance. However, despite that the global prevalence of phenotypic AMX resistance remains low, our data shows high rate of genotypic hits and more phenotypic resistance could reflect the high use of amoxicillin in other common infections including upper respiratory tract infections and urinary tract infections.
- Comment 2: Methodological issue (Lines 491–502, Broth Microdilution Procedure):
According to the described MIC range for all antibiotics (0.25–128 µg/mL), it appears that the breakpoint for AMX (EUCAST: 0.125 µg/mL) falls below the tested range. How was susceptibility to AMX accurately determined under these conditions? Please clarify this issue and provide appropriate justification, ideally citing reference guidelines for both the methodology and the breakpoints applied.
Response2: Thank you for this observation. We confirm that there was a typo error were we missed type 0.125 with 0.25. the error is now corrected. The reference is added.
- Comment 3: Referencing and clarity in methods:
The methods section lacks clear references for the applied MIC breakpoints and the sources for the broth microdilution procedures. Please ensure that appropriate literature is cited to support these choices (e.g., EUCAST, CLSI, or relevant).
Response3: Thank you for pointing out to this important information. We missed adding the references. Now it has been added in the methodology in the relevant sections.
- Comment 4: Minor point – formatting and citation style:
Several references are cited using bracketed numbers, but some are inconsistently formatted. Please check and standardize all in-text citations and references according to the journal’s guidelines.
Response4: A referencing software (Mendeley) has been used to insure consistency in the text and all references and brackets were revised.